# Explainable classification of goat vocalizations using convolutional neural networks

**Stavros Ntalampiras**[1,2¤☯*], **Gabriele Pesando Gamacchio**[1¤☯]

**1** Department of Computer Science, University of Milan, Milan, Italy, **2** Research Center on AI for Animal Health and Welfare, University of Milan, Milan, Italy

¤ Current address: Department of Computer Science, University of Milan, Milan, Italy
☯ These authors contributed equally to this work.
* stavros.ntalampiras@unimi.it

## Abstract

Efficient precision livestock farming relies on having timely access to data and information that accurately describes both the animals and their surrounding environment. This paper advances classification of goat vocalizations leveraging a publicly available dataset recorded at diverse farms breeding different species. We developed a Convolutional Neural Network (CNN) architecture tailored for classifying goat vocalizations, yielding an average classification rate of 95.8% in discriminating various goat emotional states. To this end, we suitably augmented the existing dataset using pitch shifting and time stretching techniques boosting the robustness of the trained model. After thoroughly demonstrating the superiority of the designed architecture over the contrasting approaches, we provide insights into the underlying mechanisms governing the proposed CNN by carrying out an extensive interpretation study. More specifically, we conducted an explainability analysis to identify the time-frequency content within goat vocalisations that significantly impacts the classification process. Such an XAI-driven validation not only provides transparency in the decision-making process of the CNN model but also sheds light on the acoustic features crucial for distinguishing the considered classes. Last but not least, the proposed solution encompasses an interactive scheme able to provide valuable information to animal scientists regarding the analysis performed by the model highlighting the distinctive components of the considered goat vocalizations. Our findings underline the effectiveness of data augmentation techniques in bolstering classification accuracy and highlight the significance of leveraging XAI methodologies for validating and interpreting complex machine learning models applied to animal vocalizations.

## 1 Introduction

During the last decades, there is a constantly increasing public interest in animal welfare requiring farmers to reach continuously higher animal welfare standards [1,2]. Even though there is no unique definition of animal welfare, the relevance of the affective aspects, including mood and emotions, is widely accepted [3,4]. The affective states are characterized by

**Data availability statement:** The dataset employed in this work is publicly available at https://zenodo.org/records/7530401.

**Funding:** This work was supported by the project "Advanced Methods for Sound and Music Computing" of the University of Milan. The authors acknowledge support from the University of Milan through the APC initiative. There was no external funding received for this study.

**Competing interests:** The authors declare that no competing interests exist.

short duration, albeit intense manifestations evoked by a particular object and/or event [5]. Such manifestations are coupled to behavioural decisions aiming at survival and reproduction by simultaneously seeking reward(s) and avoiding punishment [6]. Unfortunately, the present literature does not include approaches developing and/or modeling positive/negative emotional states exhibited by farm animals, let alone integrating such tools and methodologies into on-site evaluation protocols [7–10]. Animal welfare considers a series of factors, such as proper housing, nutrition, disease prevention and treatment, responsible care, etc.

Interestingly, in the recent years, the application of AI-based techniques for the analysis and understanding of animal vocalizations has garnered significant interest [11–13]. The objectives of these studies are multifarious, ranging from industrial applications such as monitoring animals within farms to assisting veterinarians in obtaining a better understanding of animal diseases [14], while several algorithms and tools have been developed to aid researchers in this area; one such example is the BIRDNET app, which focuses on identifying bird species from their vocalizations [15]. Unfortunately, to the best of our knowledge, automatic acoustic monitoring of goat farms has not been systematically explored in the related literature. Existing works approach the specific problem for other animals, e.g. identifying rale sounds in chickens [16], controlling pig farms [17], monitoring fowl farms [18], sheep farms [19], etc. Nonetheless, there is research processing goat vocalizations. Ruiz-Miranda et al. [20] studied the physical characteristics of vocalizations produced by domestic goats in response to their offspring's cries. Briefer et al. [21] presented an analysis on whether contact calls were affected by social environment and kinship during early ontogeny in goats. Interestingly, the authors of [22] showed that goats are able to distinguish between positive and negative emotion-linked vocalizations, thus they may be used for animal welfare monitoring [23].

The above-mentioned literature gap may be attributed to the limited availability of a thorough dataset of goat vocalization since publicly-available datasets are practically nonexistent. To cover this gap, the VOCAPRA project implemented such a dataset with the ultimate goal being the real-time provision of data, information and insights to farmers, animal scientists, technicians, etc. The VOCAPRA project has collected a valuable dataset of goat vocalizations from different farms located in Lombardy, Italy. The dataset includes recordings of both individual and groups of goats which have been annotated by experienced animal scientists as detailed in [23].

Based on the VOCAPRA dataset, this work proposes a novel deep neural architecture for classifying goat vocalizations. The main novelties are the following:

- design and implementation of a Convolutional Neural Network (CNN) tailored to the classification of goat vocalizations,
- utilization of a standardized feature set eliminating the necessity of domain expertise,
- consideration and application of suitable data augmentation techniques during training the CNNs towards improving the generalization capabilities of the deep architectures, while preventing overfitting,
- put in place a thorough process able to provide transparent explanations of the model's predictions,
- carry out a thorough comparative interpretation analysis highlighting the time-frequency components contributing to the excellent classification accuracy offered by the proposed CNN, and
- design a question-and-answer framework able to meaningfully interact with animal scientists and assist them in gaining insights when working with goat vocalizations.

Importantly, in order to gain insights into the underlying mechanisms of the proposed model and to identify which time-frequency components play a key role in the classification process, we employed eXplainable Artificial Intelligence (XAI) techniques. Specifically, we adapted the Concept Relevance Propagation algorithm [24] to the present problem and located the audio structures which were systematically employed by the model during classification. CRP extends the functionality of back-propagation by introducing the concept of conditions enabling the identification of particular channels and layers that have a significant impact on the final prediction. This XAI-driven approach provided clarity on the importance of specific frequency components, thus enhancing the model's interpretability. After extensive experiments, we demonstrate the superiority of the proposed solution over existing solutions. Overall, this paper seeks to contribute to the burgeoning field of animal vocalization recognition and provide insights into the potential of deep learning techniques, while emphasizing the importance of prediction interpretability.

The rest of this work is organised as follows: Sect 2 formalizes the present problem. Sect 3 provides a brief description of the available dataset, while Sect 4 explains in detail the proposed modeling framework. Subsequently, Sect 5 extensively details the experimental set-up along with the achieved results and Sects 6 and 7 shows how such predictions can be interpreted and meaningfully presented to human experts. Finally, in Sect 8 we draw our conclusions and outline future research directions.

## 2 Problem formalization

This present article assumes availability of corpus $T^s$ encompassing audio recordings characterizing goat vocalizations belonging to the following classes, i.e. class dictionary $\mathcal{D}$={*heat*, *feed distribution*, *parturition*, *injury/death*, *social isolation*, *simultaneous presence of mothers and kids*, *presence of unknown visitors*, *mother-kid separation*}. Moreover, as per the audio pattern recognition literature [25], we assume that such audio patterns associated with specific goat vocalizations follow consistent, yet unknown probability density functions denoted as $P_i, i \in \mathcal{D}$. We further assume that at each time instance, there is one dominating audio class. The final goal is to predict the class of unknown goat vocalizations following a species-independent experimental protocol.

## 3 The VOCAPRA dataset

This section briefly describes the VOCAPRA dataset, which is publicly available for research purposes at https://zenodo.org/records/7530401. It comes from the monitoring of goat farms on a 24/7 basis. More specifically, goat vocalizations were captured in the following goat farms located in the Northern Italy:

- Farm A: it comprises an organic certified farm breeding 105 lactating Alpine goats,
- Farm B: it is an intensive farm with approximately 64 lactating Alpine goats,
- Farm C: it breeds 95 lactating Saanen goats in both conventional and intensive ways, and
- Farm D: conventional intensive farm belonging to the University of Milan hosting 29 Alpine goats.

The VOCAPRA dataset encompasses a diverse array of emotional states experienced by goats in response to specific stimuli and situations encountered in their daily lives. These states are categorized into eight distinct classes:

a) **Heat**: Reflecting the physiological and behavioral changes associated with reproductive cycles and mating.

b) **Feed Distribution**: Capturing vocalizations elicited during feeding times, shedding light on social dynamics and resource allocation within goat herds.

c) **Injury or Death**: Documenting vocal expressions in response to injuries or mortality events, offering insights into distress signals and herd reactions to adverse events.

d) **Social Isolation**: Unveiling vocal cues associated with separation from herd members, highlighting the significance of social bonds and group cohesion among goats.

e) **Mother-Kid Separation**: Observing vocalizations arising from the separation of mother goats and their offspring, elucidating maternal instincts and bonding dynamics.

f) **Parturition**: Documenting vocalizations during the birthing process, providing crucial insights into maternal behaviors and birthing rituals among goats.

g) **Presence of Unknown Visitors**: Recording responses to unfamiliar human or animal presence, unveiling vigilance and territorial behaviors in farm settings.

h) **Mother-Kid Reunion**: Capturing vocal interactions upon reunion between mother goats and their kids, illuminating the significance of maternal bonds and social reunification in goat herds.

By delineating these distinct emotional states and situational contexts, the VOCAPRA dataset serves as a valuable resource for researchers seeking to unravel the complexities of goat communication and behavior in diverse farming environments.

Initial data exploration revealed an imbalanced distribution of the classes (see Fig 3), which was related to the occurrence frequency of the respective events and was taken into consideration during the model learning process. Preprocessing of the dataset involved standardizing the duration of each goat audio sample to a consistent 2-second interval using zero-padding where necessary. To gain a high-level understanding of the data, in Fig 1 we illustrate log Mel spectrograms extracted from representative samples from each class.

There, we may observe interesting differences in the time-frequency content among the various classes. For example, vocalizations associated with *mother-kid separation* exhibit significant energy in lower frequency regions with respect to *heat* vocalizations. In addition the duration of vocalizations produced during *social isolation* last longer that the ones associated with *injury or death*.

For more information regarding the recording campaign and dataset composition, the interested reader is referred to [23].

## 4 The proposed solution

In this section, we detail the proposed solution with specific focus on the model learning and classification processes. The description is divided in 1. feature extraction, 2. data augmentation, and 3. model architecture.

### 4.1 Feature extraction

In the present study, the log-Mel spectrogram was chosen as the input to the proposed CNN model. Such a feature has been found to be effective in a wide variety of speech and generalized sound recognition tasks [26,27] including computational bioacoustics applications [13,28]. The log-Mel spectrogram was extracted using the following parameters: sample rate of 22050 Hz, 128 Mel-bands, window size equal to 1024 samples with a 50% overlap between subsequent windows [29]. The extraction process was performed on each audio sample, which was first split into 1-second duration with a sliding window of 50% overlap. Unlike handcrafted features, such a standardized feature extraction process minimizes the need for

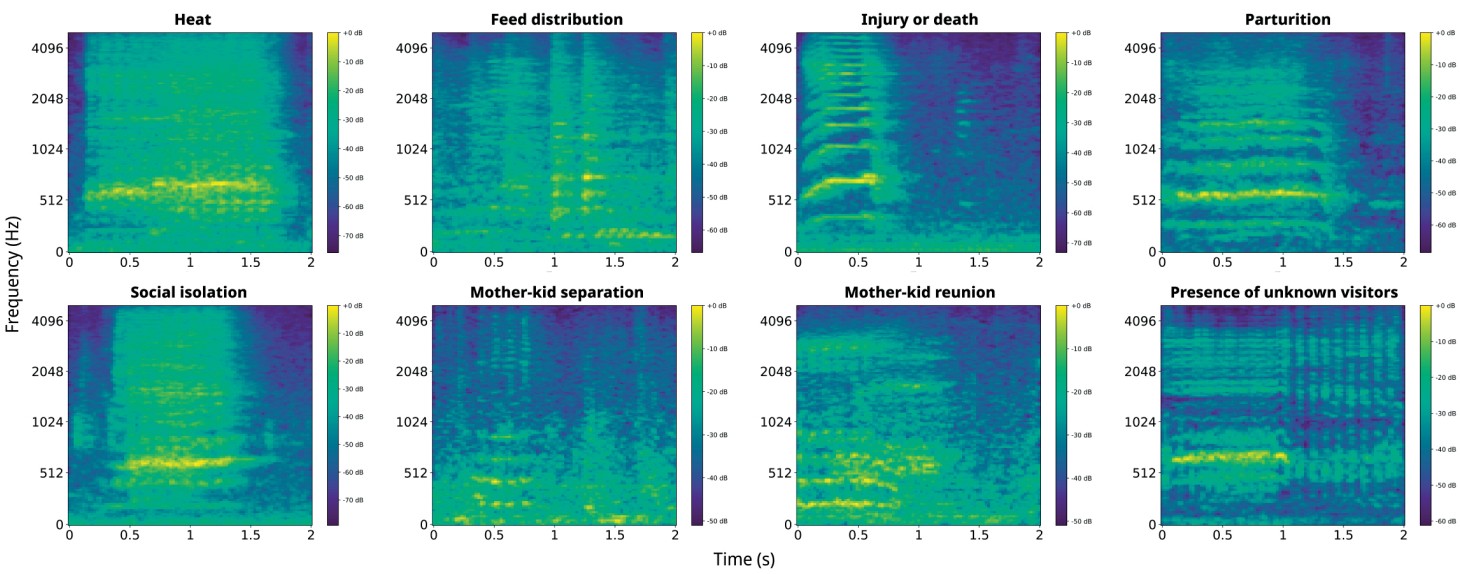

**Fig 1. Log-mel spectrograms extracted from goat vocalization representative of each available class.**

domain expertise for characterizing the available goat vocalizations. To further increase the diversity of the data set, various data augmentation techniques were applied as described next.

## 4.2 Data augmentation

CNNs, characterized by their substantial model capacity and large number of parameters, heavily rely on access to extensive training data. This dependency is crucial for effectively learning a non-linear function from input to output, facilitating generalization and achieving high classification accuracy on previously unseen data. Thus, we suggest employing audio data augmentation as a solution to address the challenge of limited data availability. Additionally, we investigate the way various augmentations impact the performance of the proposed CNN architecture. The methods used in this study include *pitch shifting* and *time stretching*, which have been successfully applied in the audio pattern recognition literature [30]. For each original audio sample, pitch shifting was applied with parameters ranging from -2 to 2 in semitones, which were expressed as [-2, 1, 0, 1, 2]. Additionally, time stretching was carried out at rates of 0.8, 1.0, and 1.2. In particular, we generated three time-stretched samples for every one-second audio sample, and for each time-stretched sample, we applied five different pitch shift values. As a result, a total of 15 log mel spectrograms were generated for each one-second audio sample. By increasing the number of samples, we obtained a larger and more diverse dataset used to train the proposed model. The augmentation parameters were selected ensuring that the identity of the goat vocalization class would not be altered up to the extent that a human listener can assess. It should be mentioned that model testing is carried out only on original data without considering their augmented versions [31].

## 4.3 The proposed convolutional architecture

CNNs have effectively addressed a wide variety of audio pattern recognition tasks [27, 32], including the computational bioacoustics [13]. Thus, this work designs a CNN

accommodating classification of goat vocalizations. Commencing with the conventional multilayer perceptron model, a CNN undergoes straightforward yet significant adjustments that empower it to efficiently handle audio signals [27]. Typically, a CNN is structured with several layers stacked together, hence creating a deep topology. The convolutional layers arrange the hidden units in a way that exposes local structures on the 2-dimensional plane, which are then utilized. This is achieved by linking each hidden unit to a small segment, known as a *receptive field*, of the input spectrogram. The units' weights act as filters, also known as *convolutional kernels*, which are employed across the entire input plane to extract a feature map. The key assumption here is that locally extracted features are valuable in different regions of the input plane, leading to the application of the same weights throughout the spectrogram. This assumption is crucial as it not only reduces the number of trainable parameters but also ensures that the network is unaffected by translational shifts in the input data [33]. Last but not least, due to the potentially vast resulting dimensionality, max-pooling layers are introduced to retain only the maximum value within a specific region.

In addressing the present task, we were not able to directly employ existing CNN architectures as the extracted log-mel spectrogram images might not necessarily fit the required input shape [34]. Importantly, resizing such inputs would cause degradation of the time-frequency content and even add undesired information. To avoid such loss of information and maintain the original feature shapes, a novel CNN architecture is proposed based on extensive empirical experimentations and inspired by the VGGish model which has been shown to be effective in audio classification tasks [35].

The proposed CNN structure is illustrated in Fig 2 and is composed as follows: there are 4 blocks of convolutional and max pooling layers. The first three blocks each contain 2 convolutional layers with 64, 128, and 256 filters, respectively, followed by a max-pooling one. The final block contains 2 convolutional layers with 512 filters each, followed by a max-pooling layer. These blocks of convolutional layers increase the depth of the neural network and allow it to learn complex and abstract features. We utilized Rectified Linear Units, i.e. the activation function is $f(x) = max(0, x)$; they have been particularly popular since they provide *a*) quicker gradient propagation compared to traditional units (such as logistic sigmoid, hyperbolic tangent, etc.), *b*) biological plausibility, and *c*) an activation form distinguished by high sparsity [36]. After the last max-pooling layer, the output is flattened into a 1D vector and passed through two fully connected layers each one encompassing 4096 units, followed by a

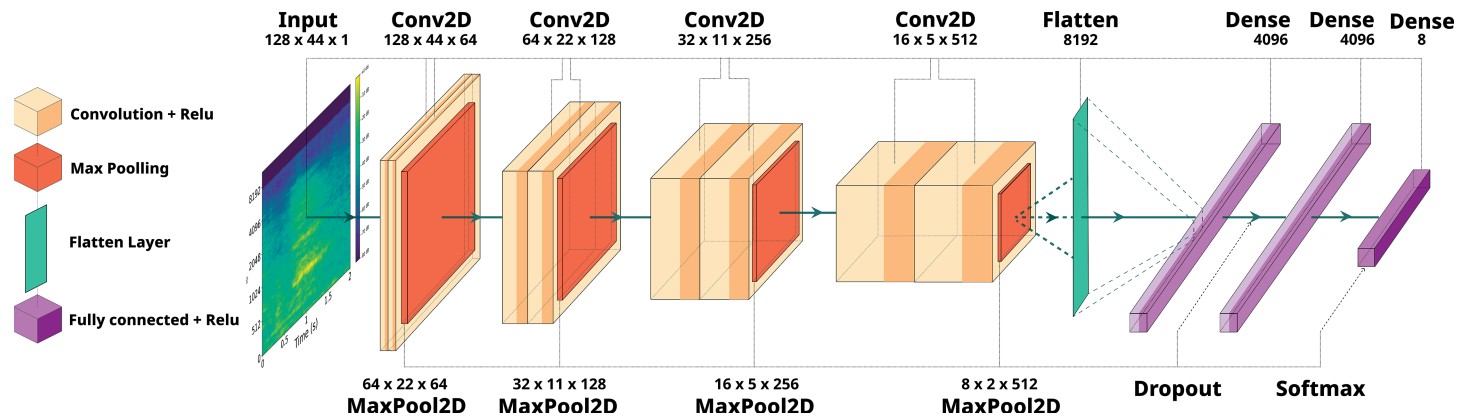

**Fig 2. The proposed model architecture composed of stacked convolutional layers for classifying goat vocalizations.**

dropout layer to remove irrelevant relationships and prevent overfitting. Finally, the output layer consists of 8 units with softmax activation, which is suitable for multi-class classification.

## 4.4 Model learning

The proposed CNN model was trained on the augmented dataset comprising 4161 recordings of goat vocalizations. The primary evaluation metric utilized was the classification accuracy computed on the original test set. For model compilation, Adam optimizer was chosen with a learning rate of 0.0001. A batch size of 12 was selected, and the model was trained for 14 epochs adopting early-stopping. Adam was chosen due to its efficiency in optimizing complex models [37]. The low learning rate value was chosen to ensure stable model convergence, while the batch size was determined based on available memory. To address class imbalance, we adopted a stratified 5-fold cross-validation scheme. As regards to the overfitting phenomenon, we monitored training and validation accuracy as well as the respective losses, where we confirmed a steady mutual increase indicating improved performance after each iteration.

## 5 Experimental results and analysis

This section presents the contrasted approaches along with the obtained experimental results following the stratified 5-fold cross validation protocol.

### 5.1 The contrasted approaches

To the best of our knowledge, this is the first time that automatic classification of goat vocalizations is addressed, hence there are no existing approaches to compare to. As such, the proposed CNN was compared to two diverse and popular deep architectures [38], namely MobileNet [34] and EfficientNet [39], which are explained next. The employed structures are depicted in Fig 4.

**5.1.1 MobileNetV2.** To expedite the classification process, we opted for transfer learning [40] using the pretrained MobileNet V2, leveraging its initial weights trained on the ImageNet dataset. This approach benefits from the knowledge captured by MobileNet while avoiding the need to train the neural network from scratch. In addition to the pre-trained layers, we added two dense layers and a final classification layer to fine-tune the network for the task at hand.

**5.1.2 EfficientNetB0.** Following an analogous line of thought, we added two dense layers with dimension 4096.

**5.1.3 Learning MobileNet and EfficientNet.** Both MobileNet and EfficientNet models were trained using the available augmented dataset, while the classification accuracy over the original test set served as the primary evaluation metric. During the training phase, we employed a batch size equal to 64 and 100 epochs. Following early experimentations, the learning rate was optimized to 0.0001.

### 5.2 Results

Fig 5 presents the confusion matrix assessing the performance of the proposed and contrasted approaches. The matrix includes average classification rates over the 5 folds with respect to each class present in the test set.

There, we observe that the proposed approach surpasses the contrasted ones in classifying all considered classes of goat vocalizations. The average rates as regards to the considered models are 95.8%, 75%, 68.3% for the proposed, EfficientNet, and MobileNet based approaches respectively. Importantly, all classes are recognized with a rate higher or equal to

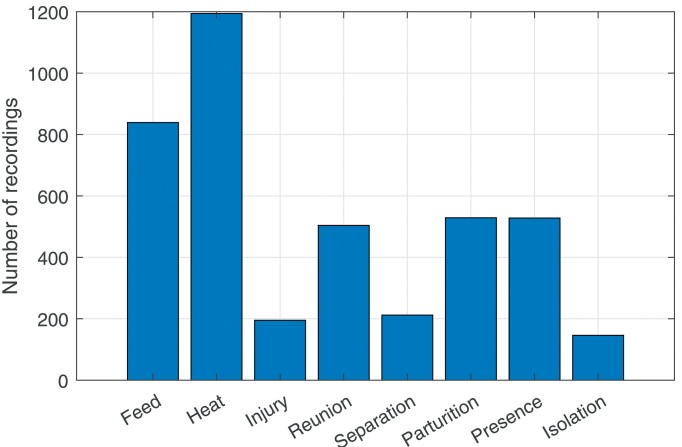

**Fig 3. Class-wise recordings' distribution of the VOCAPRA dataset.**

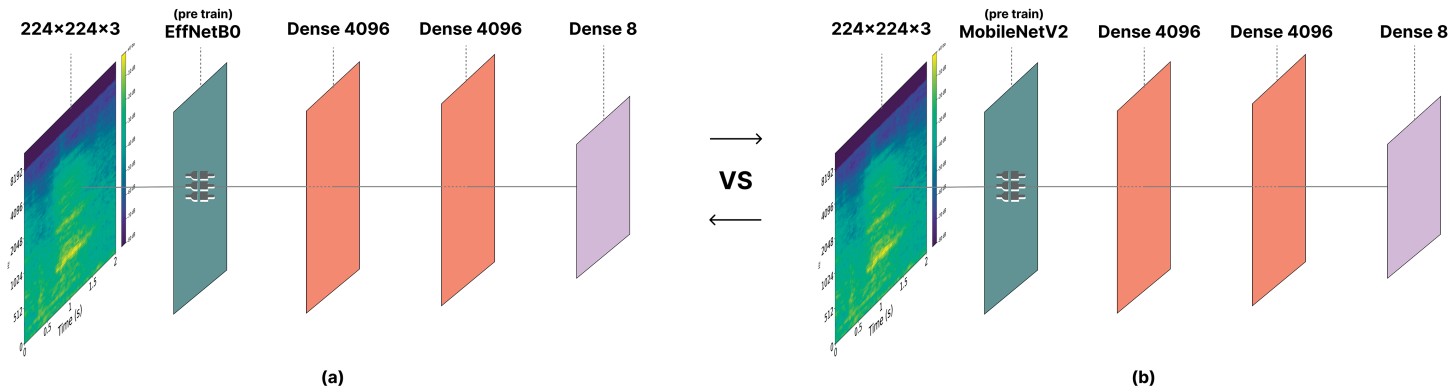

**Fig 4. Illustration of the structures characterizing the contrasted approaches, namely (a) EfficientNetB0 and (b) MobileNetV2.**

91%, demonstrating promising performance. The class recognized with the highest rate is the *isolation* one which presents a quite characteristic pattern as we see in Fig 1. It is followed by *heat*, *visitors*, *injury-death*, and *reunion*.

In addition, the rates reveal that classes *separation* and *parturition* consistently demonstrated relatively lower rates compared to the rest of the classes. This observation could be attributed to the smaller number of available audio samples in these classes, as well as to the acoustic similarity of those vocalizations with other classes to the extent that the human listener can judge. For instance, presence of unknown visitors was often misclassified as class feed distribution, possibly indicating that the goats experienced similar emotions or stimuli during these situations.

Furthermore, examination of the confusion matrix reveals that neither MobileNet nor EfficientNet models attained performance levels comparable with the proposed CNN model. It is worth noting that both approaches presented a tendency towards overfitting (see the implementation page).

During the conducted experiments, it became apparent that transfer learning, as applied through MobileNet and EfficientNet architectures, did not yield outcomes matching the predictive accuracy demonstrated by the proposed CNN model. Despite their pretrained weights,

| Presented | Responded | | | | | | | | | | | | | | | | | | | | | | |
|---|---|---|---|---|---|---|---|---|---|---|---|---|---|---|---|---|---|---|---|---|---|---|---|
| | Heat | | | Reunion | | | Isolation | | | Separation | | | Visitors | | | Feed Distr. | | | Injury-death | | | Parturition | | |
| Heat | 98 | 77 | 62 | 0,874 | 2,3 | 2,1 | 0,047 | 1,2 | 0,93 | 0,1812 | 1 | 0,94 | 0,284 | 3,2 | 2 | 0,48 | 6,1 | 7,1 | 0,051 | 0,13 | 0,26 | 0,215 | 1,8 | 1,6 |
| Reunion | 1,106 | 3,5 | 8,1 | 95,4 | 75 | 68 | 0,125 | 1,2 | 0,49 | 0,222 | 1,7 | 1,6 | 1,582 | 2,4 | 1,5 | 0,484 | 7,3 | 10 | 0,08 | 0,051 | 0,1 | 0,988 | 4,7 | 6,4 |
| Isolation | 0,114 | 8,3 | 18 | 0,557 | 2,5 | 4,3 | 98,4 | 95 | 93 | 0,045 | 1,2 | 0,71 | 0,29 | 2 | 1,5 | 0,181 | 5,1 | 5,8 | 0,078 | 0,17 | 0,77 | 0,158 | 3,1 | 6,2 |
| Separation | 2,84 | 7,3 | 16 | 1,94 | 9 | 12 | 0,136 | 1,3 | 1 | 91,8 | 62 | 46 | 1,026 | 5 | 3 | 3,158 | 7,1 | 13 | 0,188 | 0,17 | 0,2 | 0,718 | 5,9 | 7,2 |
| Visitors | 0,35 | 6,9 | 16 | 1,264 | 3,2 | 4,9 | 0,060 | 0,67 | 0,82 | 0,124 | 1,3 | 2,1 | 97,6 | 73 | 79 | 0,174 | 8,5 | 14 | 0,129 | 0,13 | 0,2 | 0,44 | 2,2 | 2,6 |
| Feed Distr. | 1,82 | 8,1 | 15 | 1,52 | 4,6 | 6,1 | 0,132 | 0,72 | 0,76 | 0,406 | 0,91 | 0,99 | 0,548 | 4,5 | 3,8 | 95,2 | 74 | 70 | 0,111 | 0,082 | 0,32 | 0,288 | 2,2 | 3 |
| Injury-death | 0,563 | 2 | 3,6 | 0,954 | 0,7 | 1,1 | 0,099 | 1,1 | 0,54 | 0,156 | 0,28 | 0,4 | 1,14 | 0,89 | 0,6 | 0,358 | 1,7 | 2,6 | 96,8 | 71 | 65 | 0,07 | 1,4 | 2,2 |
| Parturition | 1,34 | 3,9 | 7,3 | 2,46 | 6,3 | 7,3 | 0,090 | 1,1 | 0,79 | 0,268 | 1,3 | 2,7 | 1,566 | 1,9 | 1,8 | 0,566 | 4,7 | 7,5 | 0,068 | 0,27 | 0,42 | 93,4 | 73 | 63 |

● **CustomCNN**     ● **EfficientNetB0**     ● **MobileNet**

**Fig 5. Confusion matrix assessing the performance of the proposed and contrasted approaches (presentation format CustomCNN/EfficientNetB0/MobileNet).**

such models failed to leverage the distinctive structures present in the available goat vocalization dataset when expressed via log Mel spectrograms.

These findings underscore the subtle interaction between transfer learning and the intricacies of specific datasets and feature representations. It is evident that the MobileNet and EfficientNet models, while benefiting from prior knowledge encapsulated in pretrained weights, encountered limitations in adaptability when confronted with the distinct characteristics of goat vocalizations.

It is interesting to note that, similarly to the proposed CNN, the class which was poorly recognised is *mother-kid separation* and the one associated with the highest rates is *isolation*. This may be attributed to the distinctness of the acoustic patterns exhibited by such vocalizations in the time-frequency plane. Importantly, the implementation of the presented experimental pipeline is publicly available ensuring full reproducibility of the achieved results at https://colab.research.google.com/drive/1Xu-U3HY_FHgI11Ni35I69DdlP7jFFOLD# scrollTo=AMb1H9dgFsNp.

We argue that the rates achieved by the proposed model are more than satisfactory and highlight the efficacy of the implemented architecture. The proposed model may be efficiently employed to serve precision livestock farming applications [23]. However, applying the existing model to unseen locations/farms/scenarios may result in decreased performance depending on the similarity of the overall soundscape to the original training and testing data. As such, model finetuning and retraining using additional data representing the unknown setting should be carried out.

## 6 Interpreting the models' operation and predictions

Despite the excellent performance reached by the proposed model, its usability and acceptability highly depends on explaining the operation of the model and the way each prediction is made [41]. In addition, such information should be meaningfully presented to animal scientists. Interestingly, this article caries out a comparative explainability analysis among the considered models to better understand their operation and identify which are the most distinctive time-frequency components characterizing each class of goat vocalizations, thus offering valuable insights to the animal scientists.

## 6.1 Concept relevance propagation

To gain insights into the predictive mechanisms of the proposed model regarding the classification of spectrograms, we employed the Concept Relevance Propagation (CRP) technique [24]. Built upon the principles of Layer-wise Relevance Propagation [42], CRP introduces the concept of conditions, extending the capability of back-propagation. Interestingly, CRP allows for the identification of specific channels and layers that significantly influence the final prediction.

In the subsequent sections, we delve into the application of CRP to analyze the predictions generated by the proposed model, thus enhancing interpretability and gaining meaningful insights into the model's decision-making process.

The initial exploration of CPR was centered on uncovering pivotal features within the available goat vocalizations. Similarly to the way features like fur or dog eyes can be readily identified in a dataset of dog images, CRP allows us to discern the learned features of the model, providing a quantitative score that indicates the influence of each feature on the final prediction. However, when applied to spectrogram images, the identified features are not as immediately interpretable as those found in image data. Indeed, when analyzing spectrograms representing audio data, the identified features are more abstract, representing specific regions of the spectrogram, which may be characterized by a certain shape or more generally low, mid, or high-frequency bands in the spectrum.

## 6.2 Sample-wise comparative explainability analysis

We applied CRP in a uniform way to elucidate the predictions of the two best-performing models, i.e. the proposed CNN and EfficientNetB0. Initially, we employed CRP to gain a broad, high-level understanding of the features learned by each model. More specifically, we examined the significance of different features such as regions, shapes, patterns, and areas of varying frequency within the spectrogram. For each layer, we systematically analyzed all channels to determine their focus on distinct features within the spectrogram. Subsequently, we identified the top six channels in each layer that exert the most significant influence on the final prediction, and visualized their impact through suitable heatmaps. Figs 6 and 7 illustrate the relevance maps generated when the proposed model and EfficientNetB0 process a given input.

This preliminary investigation facilitated the understanding of the way the proposed CNN and EfficientNet models carry our predictions. As we see in Fig 6, the predictions of the proposed CNN are made by dividing the spectrogram in specific regions, such as low, mid, or

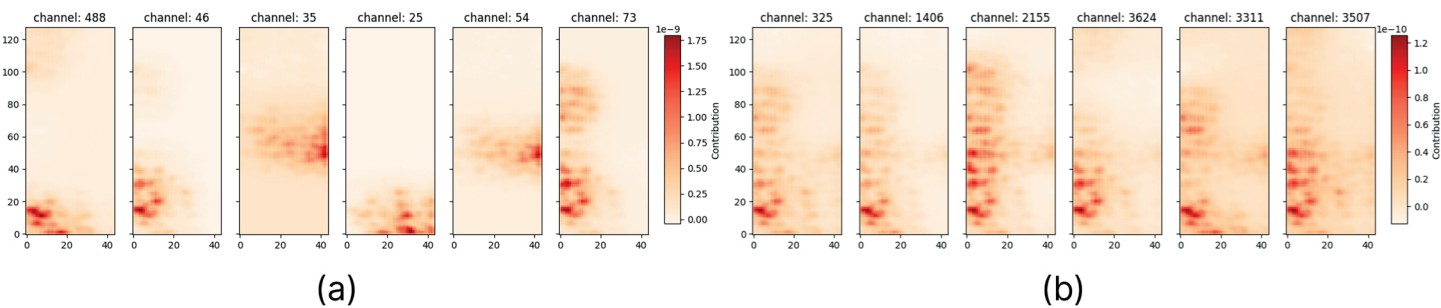

(a)  (b)

**Fig 6. CRP for a single spectrogram when processed by the proposed CNN.** (a) Heatmaps displaying the top six most relevant channels for the first fully connected layer of the proposed CNN. (b) Heatmaps illustrating the top six most relevant channels for the eighth convolutional layer in the proposed CNN.

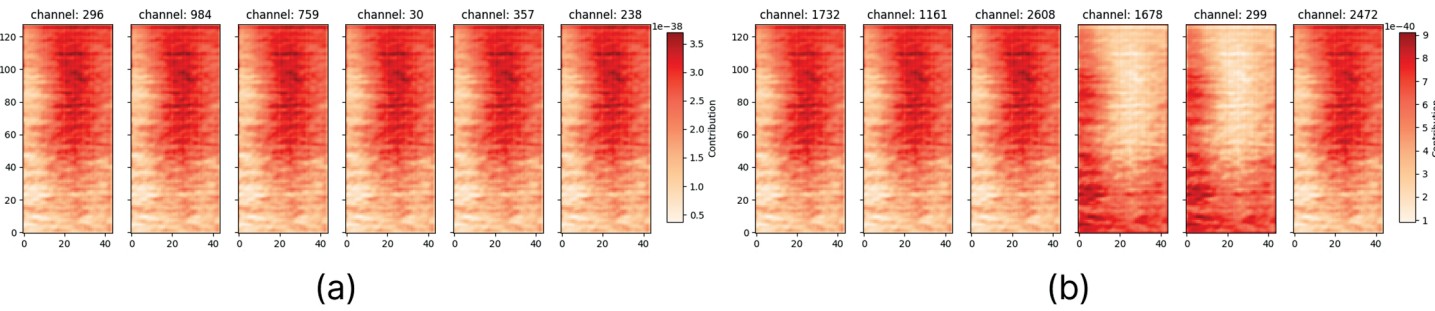

**Fig 7. CRP for a single spectrogram when processed by EfficientNetB0.** (a) Heatmaps showing the top six most relevant channels for the second fully connected layer of the EfficientNetB0 model. (b) Heatmaps representing the top six most relevant channels for the eighth convolutional layer in the EfficientNetB0 model.

high-frequency bands and considering them with diverse weights. On the contrary, the division of the spectrogram into high, mid, and low regions is not applicable to the prediction strategy of the EfficientNet model; instead, the CRP analysis suggests that the EfficientNet model exploits almost the entirety of the spectrogram using a grid approach while excluding most of the content associate with the low-frequency regions. This concise, yet comprehensive exploration has guided the subsequent phase of our investigation aiming at a class-level interpretation analysis.

## 6.3 Class-wise relevance heatmaps

The specific interpretation phase aims at offering in-depth and comprehensible insights into the way the considered models predict each goat vocalization class. Thus, we implemented the following procedure:

- selection of $n$ goat vocalizations to represent each each class as the most central ones following a $k$-means clustering scheme using the Euclidean distance.
- application of CRP analysis on each log-Mel spectrogram extracted from the selected goat vocalizations. This involved computing the relevance heatmaps of the top $g$ most relevant channels impacting the final prediction with respect to each layer.
- computation of the class-wise average heatmap by aggregating the corresponding heatmaps.

It should be noted that in the case of the proposed CNN model, the average heatmap for each class was computed using a total of 1600 relevance heatmaps. The formula used to determine the total number of relevance heatmaps used to compute the average heatmap for a single class is the following:

$$\bar{\mathcal{M}} = |\mathcal{D}| \times n \times g \qquad (1)$$

where $|D|$ denotes the cardinality of the class dictionary (see Sect 2), $g$ represents the number of relevance heatmaps per layer, and $n$ the number of spectrograms per class. In our specific case, $|D| = 8$, $g = 20$ and $n = 100$.

## 6.4 Class-wise comparative explainability analysis

The resulting class-wise average heatmaps characterizing the operation of the proposed (top row), EfficientNetB0 (middle row) and MobileNet (bottom row) models are depicted in Fig 8.

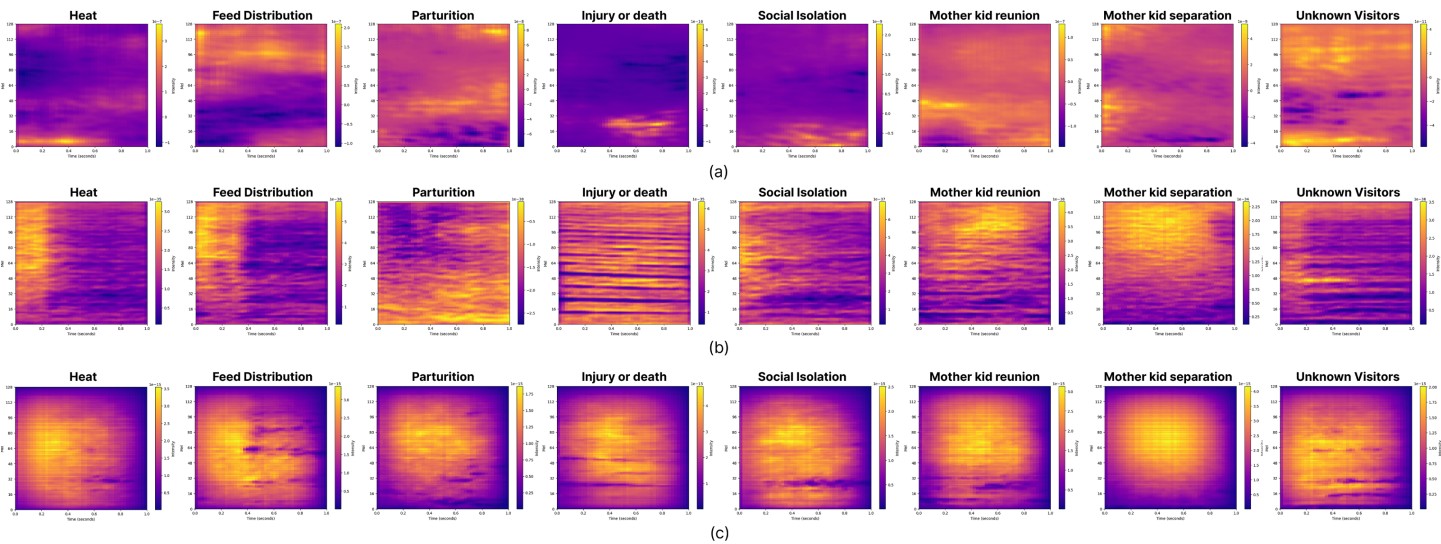

**Fig 8. The class-wise heatmaps with respect to the proposed model (top row), EfficientNetB0 (middle row), and MobileNet (bottom row).**

Importantly, there is a notable disparity in the focus areas across the various spectrogram regions, i.e.

- Custom CNN (top row): Here, the heatmaps indicate that the custom CNN learnt to place emphasis upon a specific region with respect to each class of goat vocalizations. Interestingly, the model attends to distinct frequency bands across the spectrograms. The activations are more localized, suggesting that the custom CNN has learned to focus on precise features within the spectrograms for each category. For instance: a) in "Heat" and "Feed Distribution" classes, the model focuses on the mid and lower frequency ranges. b) in "Injury or death" classes, it highlights certain diagonal patterns in the mid-range, capturing distinct features. c) "mother kid reunion" and "Unknown Visitors" show attention to broader regions, with concentration in specific time-frequency regions. Such a focused and specialized attention contributes the almost excellent performance achieved by the proposed CNN model.
- EfficientNetB0 (middle row): The respective heatmaps display broader attention regions compared to the proposed CNN. Instead of focusing on narrow bands or specific spectrogram parts, the model spreads its focus across a larger portion of the spectrogram. For example: a) in the "Heat" and "Feed Distribution" classes, the attention is more diffused, indicating that the model might struggle to pinpoint critical features as effectively as the custom CNN, b) "injury or death" and "Social Isolation" still show some distinct patterns, but the activations are less precise the ones observed by the maps characterizing the operation of the proposed model. Such a broad focus could explain why EfficientNet might not perform as well, since it captures less specific features, which are hard to detect in novel spectrograms.
- MobileNetV2 (bottom row): the heatmaps of MobileNet also show relatively diffused attention, though slightly more structured than EfficientNet. MobileNet focuses on larger frequency ranges in each audio class. For example: a) in the "Heat" and "Feed Distribution" classes, MobileNet seems to activate larger areas in the lower and mid frequencies, but the

patterns are not as crisp as in the proposed CNN. b) in "Injury or death", the model struggles to focus on the distinct diagonal pattern that the proposed CNN captures well, suggesting less precise feature extraction. Similar to EfficientNet, MobileNet's broader focus likely leads to poorer performance in classifying these specific vocalizations compared to the custom CNN.

## 6.5 Comments on model interpretability

One overarching consideration stems from the fact that CNNs are primarily designed to learn from images, rather than directly "listening" to the sounds represented by spectrograms. As evidenced by the case of the EfficientNetB0 model, while relevance maps may visually highlight certain regions, they may not necessarily correspond to the expected constituent frequencies crucial for sound classification. In several instances, these highlighted regions might have contributed to distinguishing between classes, but ultimately may not represent meaningful features from the animal scientist's point of view. Such discrepancy could be indicative of inadequate data or lack of generalization ability. Despite being trained on a vast image dataset, the need to fine-tune precomputed weights for spectrogram classification underscores the challenges in transferring knowledge from general image datasets to spectrogram-specific tasks. Spectrograms, being inherently similar across different classes, may necessitate a specialized model for accurate classification. The clarity of the average maps produced by the proposed CNN model suggests a more robust learning of relevant features, which likely contributes to its superior generalization performance.

Another important consideration is the potential utility of model prediction inspection in identifying corrupted data. For instance, by subtly embedding class labels into spectrogram images, one could assess whether the model has indeed learned to classify based on the visual cues provided by the labels. This approach enables the rapid identification of any biases or anomalies in the dataset that may affect model performance. However, it remains challenging to determine whether such analyses can effectively discern the optimal training data, as the criteria for optimal data may vary depending on the specific task and dataset characteristics. Nonetheless, the ability to detect corrupted data through model prediction inspection underscores its value in ensuring the integrity and reliability of the dataset used for training and evaluation.

## 7 Closing the usability loop

The interpretation flow described in Sect 6 may provide clear insights regarding the operation of the proposed model and the way predictions are carried out. The following step would be the usage of such information by animal scientists, i.e. closing the usability loop. Interestingly, the implemented framework presents the opportunity for animal scientists to interact with the model, allowing them to grasp the reasoning behind each prediction (see Fig 9). This interactive capability assists experts in becoming familiar with the AI-based tool, thereby bridging a notable gap highlighted in the existing literature [41,43,44] concerning the establishment of trust.

During the interaction with the classification model, the proposed framework addresses four distinct types of questions:

- identify the spectrogram components with the most notable influence on the specific decision,
- determine the goat vocalizations in $T^s$ that closely resemble the novel one,

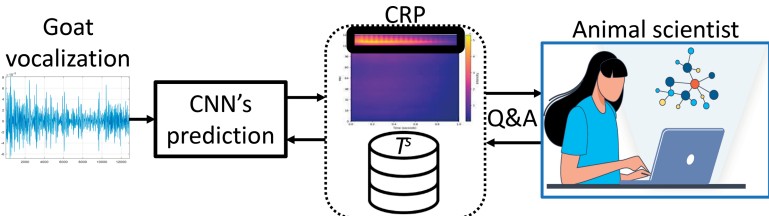

**Fig 9. Illustration of the interactive framework offering explainability insights to the animal scientists.**

- identify the goat vocalizations in $T^s$ that exhibit the most significant differences from the novel one, and
- calculate and sonify the difference(s) between the most relevant time-frequency content of two goat vocalizations.

When addressing the first question, the framework provides a comprehensive response by carrying out the next process:

a) presents the relevance map associated with the specific prediction (as for example illustrated in Fig 6),
b) directs attention to regions of heightened significance within the map,
c) pinpoints the corresponding content along the time-frequency axes, and
d) sonifies the identified content, allowing human experts to exclusively listen to that particular segment.

Concerning the second and third questions, the current solution proceeds with the calculation of the cosine distance [45] in the model embeddings space between the novel goat vocalization and the ones available in $T^s$. Thus, the system identifies those goat vocalization that are most similar or dissimilar. Notably, the expert can consistently monitor the similarity score, aiding in evaluation and providing insights into the system's confidence when assessing both similar and dissimilar vocalizations.

As regards to the fourth functionality, the system generates the relevance maps of the two goat vocalizations and subtracts them. Subsequently, the difference is projected to the spectrogram and the parts with high weights are sonified. Those signal components are the ones which heavily influenced the prediction of the model. Interestingly, this may help research and study of the specific species by highlighting differences in their vocalizations as emitted in different contexts.

We argue that such an interactive system holds the capability to substantially aid animal scientists in efficiently utilizing an AI-driven solution. The main goal is to provide explicit and easily understandable assistance to human experts. Utilizing this transparent approach guarantees a in-depth comprehension of the factors influencing the final prediction. Finally, it's worth noting that such a module aligns well with the regulations for AI-based systems which were recently released in the EU's AI Act (https://www.europarl.europa.eu/topics/en/article/20230601STO93804/eu-ai-act-first-regulation-on-artificial-intelligence).

## 8 Conclusions and future work

This article presented an automatic framework able to process goat vocalizations, interpret its predictions, and meaningfully communicate them to animal scientists. More specifically, we

explained the design and implementation of a CNN tailored to classify goat vocalizations representing eight different contexts based on the hierarchical class organisation articulated in [23] which is focused on goats' emotional states. Importantly, the proposed CNN architecture achieved excellent classification accuracy (95.8%) following a species- and farm-independent protocol. Subsequently, we illustrated how such predictions may be interpreted by employing CRP along with a comparative analysis of the considered models. It should be stressed out that the proposed model was able to discover salient features distinctively characterizing each one of the considered classes. Last but not least, the proposed framework offers the opportunity to animal scientists to interact with it gaining insights regarding its predictions, while building trust.

We argue that such a system and its potential integration with other precision livestock farming tools and technologies could enhance farm animal welfare. Nevertheless, it is crucial to acknowledge that while such technologies serve as potent tools for assisting farmers, they may pose potential threats to animal welfare [46]. At the same time, there is a potential risk to animal welfare when farmers excessively depend on these technologies, leading to a decline in their understanding of animals' needs and reduced time spent with them, resulting in adverse effects on human-animal relationship quality. Interestingly, the present framework aids in enhancing the human-goat relationship by empowering human experts to gain a better comprehension of the significance of vocalizations expressed by goats in various situations.

In the future, our objectives include:

- developing physical models that describe goat vocalizations based on insights derived from the current explainability framework,
- evaluating the performance of the proposed model across locations, farms, etc.,
- training from scratch MobileNet and EfficientNet and evaluate them as long as a sufficient amount of data becomes available,
- investigate within-species variation associated with behavioral states,
- integrating the proposed model into a smartphone application able to generate alerts as needed,
- exploring the automatic identification of individual goats potentially using few-shot learning methods, e.g. [47],
- studying the evolution of goat vocalizations over the entire period of one year using the VOCAPRA dataset, and
- systematically organizing the model and dataset to serve as educational material for young animal scientists, students, veterinarians, zootechnicians, farmers, etc.

## Acknowledgments

We would like to thank the VOCAPRA project (https://vocapra.lim.di.unimi.it/) for making available the dataset used in this work. We gratefully acknowledge the support of NVIDIA Corp. with the donation of two Titan V GPUs.

## Author contributions

**Conceptualization:** Stavros Ntalampiras.

**Data curation:** Gabriele Pesando Gamacchio.

**Formal analysis:** Stavros Ntalampiras.

**Funding acquisition:** Stavros Ntalampiras.

**Investigation:** Stavros Ntalampiras.

**Methodology:** Stavros Ntalampiras, Gabriele Pesando Gamacchio.

**Project administration:** Stavros Ntalampiras, Gabriele Pesando Gamacchio.

**Software:** Gabriele Pesando Gamacchio.

**Supervision:** Stavros Ntalampiras.

**Validation:** Stavros Ntalampiras, Gabriele Pesando Gamacchio.

**Visualization:** Gabriele Pesando Gamacchio.

**Writing – original draft:** Gabriele Pesando Gamacchio.

**Writing – review & editing:** Stavros Ntalampiras.

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
