## [Decision Letter · Decision Letter 0]

20 Sep 2024

PONE-D-24-24074Explainable Classification of Goat Vocalizations Using Convolutional Neural Networks

PLOS ONE

Dear Dr. Ntalampiras,

Thank you for submitting your manuscript to PLOS ONE. After careful consideration, we feel that it has merit but does not fully meet PLOS ONE’s publication criteria as it currently stands. Therefore, we invite you to submit a revised version of the manuscript that addresses the points raised during the review process.

**Feedback from Academic Editor:**

**Dear Authors**

**Please find reviewer feedback for your manuscript at the tail end of this email.**

**It is recommended that you address the feedback offered by the reviewers. Additionally, it is required that you address the following feedback:**

**1. Please describe the class labels i.e. [heat, feed distribution, parturition, injury/death, social isolation, simultaneous presence of mothers and kids, presence of unknown visitors, and mother-kid separation] so that the current research helps future research to be built upon it. It will be helpful if the rationale for choosing these labels is provided. Please clarify, if heat is used as a label, what temperature is considered for the label to be considered "heat"? Can two labels exist simultaneously, for example, can "heat" exist with "social isolation"? If so, which label is assigned to the recording in this case? **

**2. Please mention the version of MobileNet that was used for their experiments since multiple versions of MobileNet are available. See: https://keras.io/api/applications/mobilenet/**

**3. Please discuss and confirm that the proposed CNN architecture, EfficientNetB0, and MobileNet models were trained with the training/testing setup i.e. maximum number of epochs, learning rate, early stopping, batch size, the same seed for cross-validation etc. For the sake of balanced comparison, the same experimental setup should be used. If this is not the case, please provide adequate justification?**

**4. Given the class imbalance from Figure 3, one would think that macro-averaged f1-score is the more suitable metric to measure classification performance. In light of this, can the authors justify the use of classification accuracy as opposed to f1-score as the metric in their experiments?**

**5. In light of feedback from Reviewer 2, it is recommended that the authors provide source code to reproduce their experimental results and bring necessary context to the classification of goat vocalizations.**

**6. Please address the feedback by Reviewer 2 regarding preprocessing artifacts being identified as relevant features in your experiments.**

**7. It will be useful if the contribution of this experimental work is clarified in light of reviewer feedback.**

We look forward to receiving your revised manuscript.

Kind regards,

Zafi Sherhan Syed, PhD

Academic Editor

PLOS ONE

**Journal Requirements:**

The present work was supported by the EIP-AGRI project ``Approccio multidisciplinare per la messa a punto di un sistema di monitoraggio continuo in allevamenti caprini da latte mediante analisi delle vocalizzazioni (VOCAPRA)'' funded by the Operation 16.1.01 Cooperation of the Rural Development Programme 2014 - 2020 of Lombardy Region (Italy).

The present work was supported by the EIP-AGRI project “Approccio multidisciplinare per la messa a punto di un sistema di monitoraggio continuo in allevamenti caprini da latte mediante analisi delle vocalizzazioni (VOCAPRA)” funded by the Operation 16.1.01 Cooperation of the Rural Development Programme 2014 - 2020 of Lombardy Region (Italy). Moreover, we gratefully acknowledge the support of NVIDIA Corp. with the donation of two Titan V GPUs. 

The present work was supported by the EIP-AGRI project ``Approccio multidisciplinare per la messa a punto di un sistema di monitoraggio continuo in allevamenti caprini da latte mediante analisi delle vocalizzazioni (VOCAPRA)'' funded by the Operation 16.1.01 Cooperation of the Rural Development Programme 2014 - 2020 of Lombardy Region (Italy).

5. Please note that your Data Availability Statement is currently missing the repository name. If your manuscript is accepted for publication, you will be asked to provide these details on a very short timeline. We therefore suggest that you provide this information now, though we will not hold up the peer review process if you are unable.

Reviewers' comments:

Reviewer's Responses to Questions

**Comments to the Author**

1. Is the manuscript technically sound, and do the data support the conclusions?

Reviewer #1: Yes

Reviewer #2: No

2. Has the statistical analysis been performed appropriately and rigorously? 

Reviewer #1: Yes

Reviewer #2: N/A

3. Have the authors made all data underlying the findings in their manuscript fully available?

Reviewer #1: Yes

Reviewer #2: Yes

4. Is the manuscript presented in an intelligible fashion and written in standard English?

Reviewer #1: Yes

Reviewer #2: Yes

5. Review Comments to the Author

**Reviewer #1:** The paper titled “Explainable Classification of Goat Vocalizations Using Convolutional Neural Networks” is a wonderful attempt to focus on sounds produced by goats in different circumstances . However it claims that automatic monitoring of goat farms is not available in literature which should be written as to the best of our knowledge the automatic monitoring of goat farms is unavailable in literature and it also boasts about it being the first time automatic classification of goat vocalizations is addressed.How could this claim be justified?

The diversity of the vocapra dataset in terms of the number of farms, breeds, and environmental conditions needs to be further highlighted. The parameters for pitch shifting and time stretching selected needs to be elaborated and further what other data augmentation techniques were available and the reasons for not using them if any.

The training and validation graphs could have been added to show how much reliable the model is if possible.

In selection of classes a very important class “Predator” should have been added and analyzed.

Overall the paper is a nice contribution where authors have put forward the ideas and thoughts in a reasonable way.

**Reviewer #2: **This manuscript presents a very interesting application of supervised bioacoustic classification: recognition of goat emotional states via their vocalizations. In it’s current form, the manuscript is missing some critical details about the methodology (in particular, how testing data is selected and whether transfer learning includes training the full models or only the final layers), and from my perspective misinterprets the results of the XAI analysis. Therefore, I believe this manuscript could become a strong contribution to the literature after making three major changes: (1) clarifying and adding relevant methodological details; (2) adjusting the focus away from machine learning methodology and towards the novel application of goat emotional states; and (3) re-considering the interpretation of the XAI and re-writing the latter sections of the manuscript accordingly.

Major comments:

The abstract is vague and short on specific statements. What does it mean that your results are “notably accurate”? (instead, provide quantitative metrics such as precision and recall). What were the results of the “detailed analysis”, did you identify any specific time-frequency content that was relevant? It then mentions a “proposed framework” in the second to last sentence that wasn’t mentioned elsewhere, which “encompasses an interaction scheme” (what does it mean to encompass it, and what is the interaction scheme?). Concrete details rather than vague generalizations will make the abstract much more useful.

L75-83: what are these classes? It seems they are supposed to represent some sort of behavioral state of the goats, but they also seem like an eclectic (perhaps incomplete) and non-mutually-exclusive set of emotional states. Are there other states that you ignore? Can you justify or explain how they were chosen, and whether there is any existing literature on each of these behavioral states corresponding to specific vocalization qualities? Also, how do you determine the “true” state of the goat? Is there a “none” or “other” category for when none of these classes are dominant? Also, a minimal description of the meaning of these states is warranted - in particular, I don’t know what “feed distribution” or “parturition” mean.

Please explain the acoustic data collection of the annotated dataset: are recordings taken opportunistically with a handheld recorder? Extracted randomly, systematically, or haphazardly from continuous audio recordings? How are they trimmed and post-processed?

I believe the novelty of this work lies in the exploration and classification of goat emotional states, rather than in the machine learning bioacoustics methodologies. To date, many studies have explored variations in CNN architecture, preprocessing, and augmentation techniques, on acoustic classification performance (for a review see Stowell et al 2022 in PeerJ). Thus, the attention to very basic data augmentations (section 4.2) and two standard CNN architectures (section 4.3), all of which are already commonplace for bioacoustics ML, is not warranted in my opinion. Rather, I would encourage the authors to focus on the novel application of these techniques to an interesting problem of within-species variation associated with behavioral states. I don’t think the argument that a custom architecture is needed to accommodate the mel spectrogram shape is a strong argument: in the literature and through empirical experimentation you will find that (1) modern CNNs can handle various input sizes with little effect on performance; and (2) rescaling the input to a different shape with interpolation has negligible impact on classifier performance, so long as you don’t make it very small to the point of losing necessary resolution.

The comparison of existing and proposed model architectures is undermined by training them in different ways (different batch size, different number of epochs and stopping procedure). Why were they trained for different numbers of epochs? If you use early stopping on the existing architectures, are they perhaps less over-fit and do they perform better on the test data? Did you use saturing performance on the validation set to determine when to stop training the existing as well as proposed architectures? Additionally, it is unclear whether the “transfer learning” approach trains the full network or only the added final layers (see minor comment below) - these two scenarios fundamentally change the interpretation of the results, and training the full network is certainly necessary to test the question posed (which architecture is best?).

It is imperative to test the generalizability of models by testing them on held-out data that is not simply a random subset of the training and validation data. It is unclear in the manuscript whether the cross-validation test datasets are random splits from the training and validation data, or whether an entire data source (e.g. a separate farm) is used as a testing set for each “fold”. The latter will provide much stronger evidence of the model’s ability to generalize beyond the training samples.

I think the CRP approach is valuable here for interpreting the model’s attention to various details, I interpret the results differently from the authors’ interpretation. Specifically, the CRP is uncovering that the custom model is “cheating” by using some quirk of the preprocessed audio for classification, that is not actually relevant to the goats’ vocalizations.

L348-356: This result, and Figure 8 top row, make me highly suspicious that the model is actually using some other feature of the audio (not the goat vocalization) to distinguish the classes. For instance, there may be a quirk in the dataset by which the “heat” samples all have a particular high-pitched mechanical noise while the “mother kid reunion” lacks this noise. The consistency of these heatmaps across all classes is particularly worrying: notice that the bottom row focuses on lower frequency energy (where the goat vocalizations are actually visible in the spectrogram) and each one looks different.

More likely, now that I look at Figure 1, there is something in preprocessing that makes this high-frequency band class specific. Fig 1 shows that this high frequency band is not only above most of the goat vocalization energy, but is completely empty in the preprocessed samples (above the Nyquist frequency perhaps?). So the model seems to be picking up on some difference in this dark region, or at least that information-less region is triggering CRP. Perhaps if there was some normalization procedure that happened on a per-class basis, the value of the empty pixels corresponds to class - this is just a guess, but it is pretty clear to me that the “activation” of the CRP is on the informationless pixels at the top of the preprocessed sample.

Figure 9 does not ease these concerns: it only demonstrates that the samples have differences in this region, not that the differences pertain to goat vocalizations. If you filter the audio or crop spectrograms to this region, can you see any differences at all between classes, or even any goat vocalization signal at all? I think you will instead be mostly in the empty region of the spectrogram based on the samples illustrated in Fig 1.

The discussion and interpretation of the results in L382-397 depends on the above, and would be pretty much reversed if my suspicion is borne out.

L399-401: In fact, I think my concern here is exactly this scenario - though the authors have interpreted the model’s focus on the top of the spectrogram as a “relevant feature”, I interpret it as the CNN learning an irrelevant feature.

Here is a way you could test my hypothesis: (1) denoise the goat vocalizations, then combine them with moderate random background noise (importantly, with a high enough sample rate to extend to the top of the sample rather than only 90% of the sample height). (2) test both models’ performance on these new samples. I suspect that the EfficientNetB0 will retain some classification performance on the new samples while the custom one will get much worse, which would indicate that the custom model is paying attention to non-relevant details.

Minor comments:

L14-30 could you motivate the idea of studying goat vocalizations for animal welfare monitoring by first providing some literature/evidence that goat’s vocalizations encode their emotional state?

L30-35: This paragraph doesn’t say what kind of data the VOCAPRA project collects: recordings of individual goats? Association of specific vocalizations with behaviors? Recordings of groups of goats? Annotations of the audio recordings? Please provide some relevant details on how the data is useful for the task at hand

L39: “the problem at hand” is vague, what is the CNN designed to do? Is it a supervised classification task or some other kind of task (eg unsupervised clustering)? What are the classes or clusters it should be able to distinguish?

L41: I don’t know what you mean by a “standardized feature set” without some context

L43: specify that this the augmentation is for training the CNNs

L49: again, you haven’t described what classes you are trying to classify

L56: please provide a high-level description of what Concept Relevance Propagation does, and how it is useful to XAI

Fig 3: the classes don’t seem to map 1:1 onto the list given in L76, please clarify.

L125: in contrast to what? Providing the alternative (some sort of manual feature specification) would make this point more clear

L130-133: you could mention the large number of model parameters, which is the typical explanation for why deep learning models are prone to over-fitting

L167-173: these days CNNs can accommodate various input shapes, and also empirically, rescaling the input sample to different sizes has little effect on classifier performance

How were the training, validation, and testing datasets created? Are they from independent or overlapping sources? This has major implications for the interpretation of model performance. For instance, if testing samples are from the same source as training and validation, the model might be over-fit to the training data but still perform well on the testing data. An ideal testing set is not just a split from k-fold cross validation, but instead a truly separate set of data (e.g. data from a different set of individual goats or a different farm).

L189: did you initialize this proposed novel architecture with random weights?

L210-216: It is unclear here if you only trained the added layers and kept the feature extractor frozen with the imagenet pretrained weights, or trained the entire network. For comparison to your custom architecture, it would be critical to train the entire network. If only training the final layers, you are asking a very different question (“can we perform transfer learning without fine-tuning the feature extractor?”, rather than “which architecture is better?”), and it would not be surprising at all to see worse performance when using the frozen feature extractor.

L249-260: this interpretation depends strongly on the my previous comment

L265: the ability to apply this model beyond the studied system is unclear to me: (1) as mentioned above, it is not clear if your testing set was from a separate location compared to your training and validation set. If not, you have very little idea of how the model will generalize beyond the training data. (2) even if your testing set comes from separate locations compared to the training set, you should discuss the challenges of domain transfer here: applying the model to new locations/farms/scenarios will likely result in decreased performance depending on the similarity of the soundscape to the original training and testing data.

6. PLOS authors have the option to publish the peer review history of their article (what does this mean?). If published, this will include your full peer review and any attached files.

Reviewer #1: No

Reviewer #2: **Yes: **Sam Lapp

---

## [Author Response · Author response to Decision Letter 1]

28 Oct 2024

Thank you very much for the opportunity to improve the paper. All comments were implemented as you see in the response letter. We would like to thank the Editor and the Reviewers for their constructive comments and suggestions.

Best regards,

Stavros Ntalampiras and Gabriele Pesando Gamacchio

---

## [Decision Letter · Decision Letter 1]

27 Nov 2024

PONE-D-24-24074R1Explainable Classification of Goat Vocalizations Using Convolutional Neural NetworksPLOS ONE

Dear Dr. Ntalampiras,

Thank you for submitting your manuscript to PLOS ONE. After careful consideration, we feel that it has merit but does not fully meet PLOS ONE’s publication criteria as it currently stands. Therefore, we invite you to submit a revised version of the manuscript that addresses the points raised during the review process.

In particular, please address the following comments from reviewer 2:

The comment regarding the Training,Validation, and Test set divisions. If experiment cannot be performed at this moment, as per the suggestion of the reviewer, it may be worth adding that experiment as a limitation or future work while providing a reasonable justification why that experiment was not performed in the current manuscript.The comment regarding comparison of the "custom CNN" and pretrained models based on ImageNET, and the rationale for proposing a custom architecture. I suppose one may compare in terms of model size and parameter count, but I encourage you to provide a reasonable rationale.

We look forward to receiving your revised manuscript.

Kind regards,

Zafi Sherhan Syed, PhD

Academic Editor

PLOS ONE

Journal Requirements:

Reviewers' comments:

Reviewer's Responses to Questions

**Comments to the Author**

1. If the authors have adequately addressed your comments raised in a previous round of review and you feel that this manuscript is now acceptable for publication, you may indicate that here to bypass the “Comments to the Author” section, enter your conflict of interest statement in the “Confidential to Editor” section, and submit your "Accept" recommendation.

Reviewer #1: All comments have been addressed

Reviewer #2: (No Response)

2. Is the manuscript technically sound, and do the data support the conclusions?

Reviewer #1: Yes

Reviewer #2: Partly

3. Has the statistical analysis been performed appropriately and rigorously? 

Reviewer #1: I Don't Know

Reviewer #2: I Don't Know

4. Have the authors made all data underlying the findings in their manuscript fully available?

Reviewer #1: Yes

Reviewer #2: Yes

5. Is the manuscript presented in an intelligible fashion and written in standard English?

Reviewer #1: Yes

Reviewer #2: Yes

6. Review Comments to the Author

Reviewer #1: (No Response)

Reviewer #2: Overview:

The analysis of model interpretability is much improved, now that the authors have fixed the preprocessing artifacts to which the pre-trained models were paying too much attention. I still feel that the framing of the manuscript as a test of a custom architecture versus pre-trained architectures is not supported by the experimental methods and results. In particular, the effects of architecture (custom vs Efficientnet vs MobileNet) and training strategy (transfer learning with frozen weights, transfer learning with unfrozen weights, training from scratch) are confounded in the experiment. Please see my comments below.

Comments:

Training,Validation, and Test set divisions:

The authors assert that they could not hold out an entire location for a test set, but that their random division of samples into k-fold validation splits represents a strong test of model performance in real world conditions. I respectfully disagree. Audio data tends to be highly auto-correlated. Imagine for instance that training and test samples for a particular scenario were from two adjacent 3-second audio clips where an individual goat repeatedly made the same vocalization. The model could be highly over-fit on training samples and still perform well on the test set, but might perform very poorly on a fresh set of audio recordings from different individual goats or different dates. It should be at least possible to split out entire original audio recordings (which are then split into many fixed-length samples) into the training and testing sets, if they cannot be split out by geographic location.

15) L210-216 original comment: It is unclear here if you only trained the added layers and kept the feature extractor frozen with the imagenet pretrained weights, or trained the entire network. For comparison to your custom architecture, it would be critical to train the entire network. If only training the final layers, you are asking a very different question (“can we perform transfer learning without fine-tuning the feature extractor?”, rather than “which architecture

is better?”), and it would not be surprising at all to see worse performance when using the frozen feature extractor.

Response of the authors:

Thank you very much for this comment. As mentioned in sections 5.1.1 and 5.1.2 we employed a classical transfer learning methodology dictating the pretrained networks to be used as feature extractors, while adding layers serving the present classification task. Please note that this is the typical way transfer learning technologies are applied. In the present case, it would be impossible to retrain such deep networks on the present dataset which is much smaller in size with respect to their original training sets (e.g. imagenet) without overfitting. This is the typical way as found in the related literature [1].

Reviewer comment round 2:

It is true that it would be difficult to fine-tune an entire architecture based on a small training set, but the authors trained the “custom CNN” on the training set so I don’t understand why they couldn’t also compare performance to a variation where the feature extractor is also trained (perhaps with a lower learning rate on the feature extractor than on the added layers). The manuscript frames this experiment as a comparison of architectures, rather than a comparison of feature extractors (architecture + data + training procedure). It is fine to compare the performance of “a custom feature extractor trained on our data with architecture A” to “a pre-trained feature extractor trained on ImageNET with architecture B”, but this is not a fair comparison of the architectures A and B themselves, nor is it a fair comparison of training the feature extractor on local data versus transfer learning: both variables have changed simultaneously. If the authors do not wish to run further experiments, they should reframe the framing of the methods and interpretation of the results as "we tried a few reasonable things and our custom model worked best" but not as a rigorous test of one architecture or training approach against another.

L245: Transfer learning approaches may either freeze or continue training the pre-trained model. Please explicitly state that the pre-trained model weights were frozen during training since that is the strategy used here.

Spectrogram input shape (L197): I maintain that you can feed any size image without resizing it into the standard PyTorch implementations of CNN architectures. This is not a justification for needing a custom architecture.

7. PLOS authors have the option to publish the peer review history of their article (what does this mean?). If published, this will include your full peer review and any attached files.

Reviewer #1: No

Reviewer #2: **Yes: **Sam Lapp

---

## [Author Response · Author response to Decision Letter 2]

6 Dec 2024

Thank you very much for the opportunity to improve the paper. All comments were implemented as you see in the response letter. We would like to thank the Editor and the Reviewers for their constructive comments and suggestions.

---

## [Decision Letter · Decision Letter 2]

19 Jan 2025

Explainable Classification of Goat Vocalizations Using Convolutional Neural Networks

PONE-D-24-24074R2

Dear Dr. Ntalampiras,

We’re pleased to inform you that your manuscript has been judged scientifically suitable for publication and will be formally accepted for publication once it meets all outstanding technical requirements.

Kind regards,

Zafi Sherhan Syed, PhD

Academic Editor

PLOS ONE

Reviewers' comments:

Reviewer's Responses to Questions

**Comments to the Author**

1. If the authors have adequately addressed your comments raised in a previous round of review and you feel that this manuscript is now acceptable for publication, you may indicate that here to bypass the “Comments to the Author” section, enter your conflict of interest statement in the “Confidential to Editor” section, and submit your "Accept" recommendation.

Reviewer #1: All comments have been addressed

2. Is the manuscript technically sound, and do the data support the conclusions?

Reviewer #1: Yes

3. Has the statistical analysis been performed appropriately and rigorously? 

Reviewer #1: N/A

4. Have the authors made all data underlying the findings in their manuscript fully available?

Reviewer #1: Yes

5. Is the manuscript presented in an intelligible fashion and written in standard English?

Reviewer #1: Yes

6. Review Comments to the Author

Reviewer #1: (No Response)

7. PLOS authors have the option to publish the peer review history of their article (what does this mean?). If published, this will include your full peer review and any attached files.

Reviewer #1: No

---

## [Editor Report · Acceptance letter]

PONE-D-24-24074R2

PLOS ONE

Dear Dr. Ntalampiras,

I'm pleased to inform you that your manuscript has been deemed suitable for publication in PLOS ONE. Congratulations! Your manuscript is now being handed over to our production team.

Kind regards,

on behalf of

Dr. Zafi Sherhan Syed

Academic Editor

PLOS ONE